# Quitting patient care and career break intentions among general practitioners in South West England: findings of a census survey of general practitioners

Emily Fletcher,[1] Gary A Abel,[1] Rob Anderson,[2] Suzanne H Richards,[1] Chris Salisbury,[3] Sarah Gerard Dean,[4] Anna Sansom,[1] Fiona C Warren,[5] John L Campbell[1]

► Prepublication history and additional material is available. To view please visit the journal ()

[1]Primary Care Research Group, University of Exeter Medical School, Exeter, UK
[2]Evidence Synthesis & Modelling for Health Improvement, University of Exeter Medical School, Exeter, UK
[3]Centre for Academic Primary Care, NIHR School for Primary Care Research, School of Social and Community Medicine, University of Bristol, Bristol, UK
[4]Psychology Applied to Rehabilitation and Health, University of Exeter Medical School, University of Exeter, Exeter, UK
[5]Institute of Health Research, University of Exeter Medical School, Exeter, UK

**Correspondence to**
Professor John L Campbell;
john.campbell@exeter.ac.uk

## ABSTRACT

**Objective** Given recent concerns regarding general practitioner (GP) workforce capacity, we aimed to describe GPs' career intentions, especially those which might impact on GP workforce availability over the next 5 years.

**Design** Census survey, conducted between April and June 2016 using postal and online responses, of all GPs on the National Health Service performers list and eligible to practise in primary care. Two reminders were used as necessary.

**Setting** South West England (population 3.5 million), a region with low overall socioeconomic deprivation.

**Participants** Eligible GPs were 2248 out of 3370 (67% response rate).

**Main outcome measures** Reported likelihood of permanently leaving or reducing hours spent in direct patient care or of taking a career break within the next 5 years and present morale weighted for non-response.

**Results** Responders included 2177 GPs engaged in patient care. Of these, 863 (37% weighted, 95% CI 35% to 39%) reported a high likelihood of quitting direct patient care within the next 5 years. Overall, 1535 (70% weighted, 95% CI 68% to 72%) respondents reported a career intention that would negatively impact GP workforce capacity over the next 5 years, through permanently leaving or reducing hours spent in direct patient care, or through taking a career break. GP age was an important predictor of career intentions; sharp increases in the proportion of GPs intending to quit patient care were evident from 52 years. Only 305 (14% weighted, 95% CI 13% to 16%) reported high morale, while 1195 (54% weighted, 95% CI 52% to 56%) reported low morale. Low morale was particularly common among GP partners. Current morale strongly predicted GPs' career intentions; those with very low morale were particularly likely to report intentions to quit patient care or to take a career break.

**Conclusions** A substantial majority of GPs in South West England report low morale. Many are considering career intentions which, if implemented, would adversely impact GP workforce capacity within a short time period.

**Study registration** NIHR HS&DR - 14/196/02, UKCRN ID 20700.

## Strengths and limitations of this study

► Census survey of all general practitioners (GPs) in South West England, providing a cross-sectional overview of quit and career break intentions.
► High response rate, reflecting both rigorous planning and implementation of the survey, and interest among GPs in the subject matter of workforce challenges.
► The potential for non-response bias was reduced by weighting estimates to account for differential response rates by age, gender and practice role.
► The survey was cross-sectional, rather than longitudinal in design; previous work demonstrates that intention to quit is a strong predictor of actually leaving.
► The research was undertaken within a rapidly changing GP workforce policy and practice environment.

## INTRODUCTION

UK healthcare has traditionally had a strong primary care base in which general practitioners (GPs) have a central role in providing care across many settings, including within general practices, in out-of-hours care and in walk-in centres. Ninety per cent of UK National Health Service (NHS) patient contact takes place within primary care—1.3 million consultations every working day, 340 million consultations per year and with a projected primary care workload of 430 million consultations per year by 2018.[1] [2] Around 74% of primary care contacts take place with a GP. General practice has been described as 'the jewel in the crown' of the NHS.[3] GPs are trained and have particular abilities in the diagnosis and management of patients with complex multimorbidity.

UK general practice is, however, facing major problems regarding maintaining the

GP workforce, with imminent GP shortages and a concomitant potential risk to patient care. A near quadrupling of unfilled GP posts was observed between 2010 and 2013 (from 2.1% to 7.9%), associated with an overall reduction in the number of GPs in England from 62 per 100 000 in 2009 to 59.5 in 2013.[4] An estimated 12% of 2947 GP training places were unfilled in England in 2013/2014. These issues are compounded by an ageing GP workforce (30% of the 43 000 current GPs are over 50 years old[4]). Workforce issues are especially pertinent in inner city settings where recruitment and retention difficulties are further exacerbated by issues relating to the sociodemographic mix of the population and to increased demands for care.

Research commissioned by the British Medical Association (BMA) has highlighted the continuing recruitment and retention challenges. In a 2014 study of 431 doctors from the 2006 cohort of medical graduates, those in general practice reported the lowest morale, with higher than expected workload being identified as a key problem.[5] Of male doctors, 90% plan to work full time compared with just 40% of female doctors.[4] Younger GPs currently appear to be more reluctant to take on the financial risks and responsibilities of becoming a partner in a practice, leading to existing partners experiencing an increased workload.[6] Recent surveys and reports have suggested that many GPs are considering permanently leaving direct patient care.[1 4 7] Published evidence suggests that nearly half of GPs leaving general practice between 2009 and 2014 were under 50 years old.[8]

There is thus an urgent need to find ways of retaining the GP workforce. If unaddressed, 'meltdown' in NHS care may follow within the foreseeable future.[1] The situation has been described as a 'crisis',[9 10] and there has been a call for policies and strategies to help retain GPs in direct patient care.[4 11] The future of NHS care over the next 5 years[12] is likely to involve new models of care, with innovations involving increasing professional skill mix and new approaches to managing the service[12 13] and in establishing federations of previously independent practices.[12] It is vital that the GP workforce is sustained through this period of change.

Our research targets the critically important area of retention of experienced GPs in direct patient care, especially focussing on those GPs who may be considering retirement, and those GPs who have taken a career break (most often on account of family reasons). Following piloting of our questionnaire in 2015,[14] this paper reports the findings of a recent census survey in which we aimed to describe GPs' career intentions, especially those which might impact on GP workforce availability over the next 5 years. The survey constitutes one part of a comprehensive programme of work, seeking to identify implementable policies and strategies to support the retention of experienced GPs in direct patient care and to support the return of GPs following a career break.

## METHODS

### Study sample

The sampling frame was the Medical Performers List of all GPs registered to practise in South West England, an area with a population of 3.5 million patients. National Performers List provides reassurance for the public that GPs, dentists and opticians practising in the NHS are suitably qualified, have up-to-date training, regulatory checks and appropriate English language skills.[15] There were no exclusion criteria. The survey was administered between April and June 2016.

### Questionnaire

The questionnaire was based on that used within earlier work,[14] modified to increase the number of questions from 11 to 24, including rewording of four questions to reflect alignment of wording with other questionnaires of broadly similar intent and by providing clear definitions for key concepts in the questionnaire including a career break, taking steps towards changing work–life balance and defining a clinical session. The questionnaire (see online supplementary material) comprised items that asked GPs about their career intentions, reporting on the likelihood that they would permanently leave direct patient care within the next 2 years or within the next 5 years. GPs were also asked to report the likelihood that they would take a career break within the next 5 years or that they would reduce their weekly average hours spent in direct patient care during this time period. GPs rated the likelihood of these events from 'very likely' to 'very unlikely' using a four-point scale. The questionnaire also included a question about current level of morale and captured general demographic data: gender, age, ethnicity, region and year of graduation and current GP employment status (eg, partner, salaried), number and pattern of sessions worked in a typical week and involvement in delivering out-of-hours care.

### Data collection

GPs were sent study materials through the post either to their practice or home address, and also by email, where available. The questionnaire was available for completion by post and online. The survey was supported by a comprehensive strategy of publicising the research through routine newsletters and circulars of relevant organisations and networks, including local medical committees, clinical research networks, Health Education England South West, the Royal College of General Practitioners, University of Exeter Medical School and the South West Academic Health Science Network.

If a GP returned multiple online or postal surveys, only the first response received by the research team was analysed. Postal response data were double entered and discrepancy checking was undertaken. Response data were stored securely and without participant name or address.

## Patient involvement

Although the study participants were GPs rather than patients, patient representatives contributed to the design of the survey. The planned work was presented to the wider project's Patient and Public Involvement (PPI) group,[16] by way of sharing the process and to check the integrity of the work, and the group provided supportive feedback. The survey results were presented at a project management group meeting, which included PPI representatives who directly contributed to interpreting and contextualising the results.

## Statistical analysis

Differential response rates between different groups of GPs would potentially introduce bias into crude survey findings. To counter this, we employed non-response weights. Inverse probability weights were calculated based on three factors: age (<40, 40–49, 50–54, 55–59 and 60 years and over), gender (male and female) and role (partner, salaried and locum/other). By employing these weights, we estimated what responses we would have received with a 100% response rate under the assumption that non-responders would have responded similarly to GPs of the same age, gender and role. Logistic regression was used to investigate the association between responses to questions regarding future career intentions (permanently leaving direct patient care within the next 2 and 5 years, taking a career break within the next 5 years and reducing average hours spent in direct patient care within the next 5 years). Each of the four sets of responses was dichotomised into 'very likely' and 'likely' versus other responses. Initially, unadjusted associations were examined for effects attributable to the explanatory factors of gender, age, country of qualification, ethnicity, role/ position and rating of current morale. Subsequently, regression models adjusting simultaneously for all explanatory factors were used to examine adjusted associations. Similar models were used with reported morale as the outcome (but not including morale as an explanatory factor). Regression analyses were restricted to those respondents with complete data on gender, age, country of qualification, ethnicity, role/position and rating of current morale.

## Supplementary analysis

Interactions were explored between various factors in the models. While some of these were found to be statistically significant, the magnitude of the interaction terms was generally small and did not alter the interpretation of the data, with one exception commented on in the results. For this reason, we have not reported the more complex interaction models here.

In addition, we considered the possibility that some groups of GPs, for example female GPs, may not report the intention to reduce hours spent in direct patient care because they already work fewer hours, on average, than other groups. We therefore performed two supplementary analyses, including either a binary variable indicating part-time working (defined as working less than eight sessions per week) or a continuous variable detailing the reported number of sessions worked per week.

All analyses were performed in Stata V.14.2.

## RESULTS

Questionnaires were distributed by post to 3370 GPs, with 1841 (55%) of these GPs also being sent the questionnaire by email, where we had a valid email address (see online supplementary table 1). Completed questionnaires were received from 2248/3370 GPs who were surveyed (response rate 67%). Of the 2248 GP respondents, 673 (30%) used the online survey. Response rates were as high for both men and women (67% and 66% respectively). Participation was lower among GPs aged under 40 years (54%) when compared with the older GPs aged 50 and over (in excess of 68% in each age group) and was lower for salaried GPs (57%) when compared with GP partners (71%) and non-principal/locum GPs (64%).

The median age of respondents was 48 years (table 1; IQR 40 to 55, range 28 to 84 years). Eighty-five per cent of respondents reported having a practice with which they were primarily affiliated; 25% of respondents reported that they were involved in the delivery of out-of-hours primary medical care. The majority (62%) of respondents were partners in their practices.

## Career intentions

Of the 2248 responding GPs, 55 had already permanently left direct patient care; 16 had selected 'none of the above' to reflect their working status in direct patient care—that is, that they were not currently working in direct patient care nor on a career break, but had not permanently quit—and were removed from further analysis. Differential response by age and role meant that GP groups reporting an intention to quit direct patient care were somewhat over-represented, introducing a small bias into the crude results which is accounted for in the weighted percentages.

Of the 2177 GP participants included in the analysis, 473 (weighted percentage 20.3%, 95% CI 18.7% to 22.0%) reported a high likelihood of quitting direct patient care ('likely' or 'very likely') within the next 2 years and 863 (weighted percentage 36.8%, 95% CI 34.8% to 38.8%) within the next 5 years (table 2). There were 1252 (weighted percentage 56.7%, 95% CI 54.6% to 58.8%) participants who reported being likely or very likely to reduce hours, and 770 (weighted percentage 36.3%, 95% CI 34.3% to 38.4%) who reported the intention to take a career break within the next 5 years. Considered together, 1535 (weighted percentage 70.0%, 95% CI 68.0% to 71.9%) participants reported that they were likely/very likely to pursue a career intention (one or more of the four presented) that would potentially adversely impact the workforce available in South West England within the next 5 years. The majority of participants also had low morale, with a substantially greater

**Table 1** Characteristics of responding general practitioners (GPs; n=2248)

| | Proportion; n (%) |
|---|---|
| **Gender** | |
| Male | 1053 (46.8) |
| Female | 1190 (52.9) |
| Prefer not to say | 3 (0.1) |
| Missing | 2 (0.1) |
| **Age (years)** | |
| Under 40 | 497 (22.1) |
| 40–49 | 735 (32.7) |
| 50–54 | 394 (17.5) |
| 55–59 | 408 (18.2) |
| 60 or over | 209 (9.3) |
| Missing | 4 (0.2) |
| Spoiled | 1 (0.0) |
| **Ethnic group** | |
| White | 2100 (93.4) |
| Mixed/multiple ethnic groups | 29 (1.3) |
| Asian/Asian British | 78 (3.5) |
| Black/African/Caribbean/ Black British | 9 (0.4) |
| Other ethnic group | 19 (0.9) |
| Missing | 13 (0.6) |
| **Region of qualification** | |
| UK/Ireland | 2107 (93.7) |
| Europe (non-UK/Ireland) | 70 (3.1) |
| South Asia | 21 (0.9) |
| Other | 42 (1.9) |
| Missing | 7 (0.3) |
| Spoiled | 1 (0.0) |
| **Role/position** | |
| GP partner | 1403 (62.4) |
| Salaried GP | 454 (20.2) |
| Locum GP | 287 (12.8) |
| Other | 68 (3.0) |
| Missing | 7 (0.3) |
| Spoiled | 29 (1.3) |

proportion of participants (1195; weighted percentage 54.4%, 95% CI 52.3% to 56.5%) reporting 'low' or 'very low' levels of morale compared with only 305 (weighted percentage 14.2%, 95% CI 12.8% to 15.8%) reporting 'high' or 'very high' morale.

## Associations between GP characteristics and career intentions/morale

Figure 1 illustrates the reported intention to quit direct patient care of responding GPs broken down by gender

and each year of GP age, aggregating likely and very likely responses together. Reported quit intentions are strongly related to age, remaining low (<20%) among younger GPs (45 years and under) for quitting direct patient care in both of the next 2 and 5 years. Both of these outcomes show a sharp rise from the age of 52 years, with the proportion stating they were likely to quit in 5 years rising to almost 90% by the age of 56 years. The proportion anticipating taking a career break as 'likely' was highest for the youngest GPs, especially among younger women where nearly 9 out of 10 female GPs aged 30 years reported the intention to take a career break, presumably on account of anticipated maternity leave.

Of the 2177 respondents reporting that their current role involved direct patient care, 2119 (97%) provided complete information on gender, age, region of qualification, ethnicity, role/position and rating of current morale and were included in the regression analyses. Tables 3 and 4 show the results of the unadjusted and adjusted logistic regression analyses, respectively, for career intentions. In unadjusted analyses, there is strong evidence (p<0.001 for all) that gender and role are associated with being likely to report the intention to leave patient care within both 2 and 5 years, with female GPs being less likely to quit patient care than male GPs (OR 0.55, 95% CI 0.46 to 0.66 for leaving patient care in 5 years) and with locum GPs being the most likely to quit patient care and salaried GPs least likely. However, these associations are likely be confounded, for example by the fact younger doctors were more likely to be female compared with older doctors. Once adjustment is made for all other factors, the effects of these characteristics on intentions to quit are no longer significant. However, after adjustment for other variables, women were still substantially less likely to report intending to reduce hours spent in direct patient care (OR 0.58, 95% CI 0.47 to 0.71), and locum GPs remained more likely to report intending to take a career break when compared with GPs in other roles.

Age was a very strong predictor of reported intention to leave direct patient care, a finding which persisted after adjustment (for example 60–69 vs 40–49 year olds OR=194.4 95% CI 84.3 to 448.3 for leaving patient care in five years). Age was also a strong predictor of the intention to reduce hours in the adjusted model, with older GPs more likely to report this intention than younger GPs. Female GPs, GPs aged under 40 years and locum GPs were most likely to report the intention to take a career break after adjustment for other factors. A model including an interaction between age and gender showed that the effect of under 40 year olds being more likely to report intentions to take a career break was strongest in female GPs, consistent with the pattern shown in figure 1 (results not shown).

Self-reported morale was a strong predictor of all four outcomes shown in tables 2 and 3. In the unadjusted analysis, a 'U-shaped' relationship is seen for all four quitting outcomes, with those with 'very high' morale more likely

**Table 2** Career intentions of responding general practitioners (n=2177)

| | How likely is it that you will permanently leave direct patient care within the next 2 years? | | | How likely is it that you will permanently leave direct patient care within the next 5 years? | | | How likely is it that you will reduce your weekly average hours spent in direct patient care within the next 5 years? | | | How likely is it that you will take a career break (or another career break) within the next 5 years? | | |
|---|---|---|---|---|---|---|---|---|---|---|---|---|
| | n | % crude | % weighted (95% CI) | n | % crude | % weighted (95% CI) | n | % crude | % weighted (95% CI) | n | % crude | % weighted (95% CI) |
| Very likely | 255 | 11.7 | 10.8 (9.6 to 12.1) | 607 | 27.9 | 25.4 (23.6 to 27.2) | 773 | 35.5 | 34.0 (32.1 to 36.1) | 453 | 20.8 | 20.7 (19.1 to 22.5) |
| Likely | 218 | 10.0 | 9.5 (8.4 to 10.8) | 256 | 11.8 | 11.4 (10.1 to 12.8) | 479 | 22.0 | 22.7 (20.9 to 24.5) | 317 | 14.6 | 15.6 (14.0 to 17.2) |
| Unlikely | 795 | 36.5 | 36.6 (34.6 to 38.7) | 675 | 31.0 | 32.3 (30.3 to 34.3) | 585 | 26.9 | 27.6 (25.7 to 29.5) | 756 | 34.7 | 34.9 (32.9 to 36.9) |
| Very unlikely | 899 | 41.3 | 42.6 (40.5 to 44.8) | 622 | 28.6 | 30.2 (28.2 to 32.2) | 326 | 15.0 | 15.2 (13.7 to 16.8) | 614 | 28.2 | 27.3 (25.5 to 29.2) |
| Missing | 10 | 0.5 | 0.5 (0.2 to 0.9) | 15 | 0.7 | 0.7 (0.4 to 1.2) | 13 | 0.6 | 0.5 (0.3 to 0.9) | 34 | 1.6 | 1.4 (1.0 to 2.0) |
| Spoiled | 0 | 0.0 | 0.0 | 2 | 0.1 | 0.1 (0.0 to 0.4) | 1 | 0.1 | 0.0 (0.0 to 0.2) | 3 | 0.1 | 0.1 (0.0 to 0.4) |
| **How would you describe your current level of morale?** | | | | | | | | | | | | |
| Very low | 352 | 16.2 | 15.6 (14.2 to 17.2) | | | | | | | | | |
| Low | 843 | 38.7 | 38.8 (36.7 to 40.9) | | | | | | | | | |
| Neither low nor high | 664 | 30.5 | 30.8 (28.9 to 32.8) | | | | | | | | | |
| High | 276 | 12.7 | 12.9 (11.6 to 14.5) | | | | | | | | | |
| Very high | 29 | 1.3 | 1.3 (0.9 to 1.8) | | | | | | | | | |
| Missing | 9 | 0.4 | 0.4 (0.1 to 0.5) | | | | | | | | | |
| Spoiled | 4 | 0.2 | 0.2 (0.1 to 0.8) | | | | | | | | | |

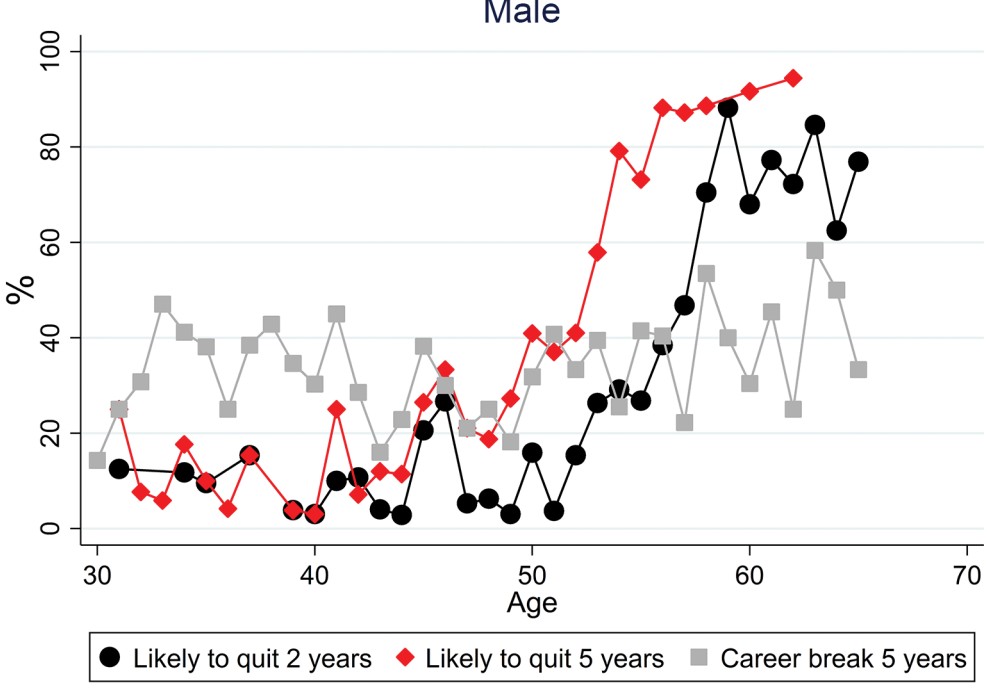

Male

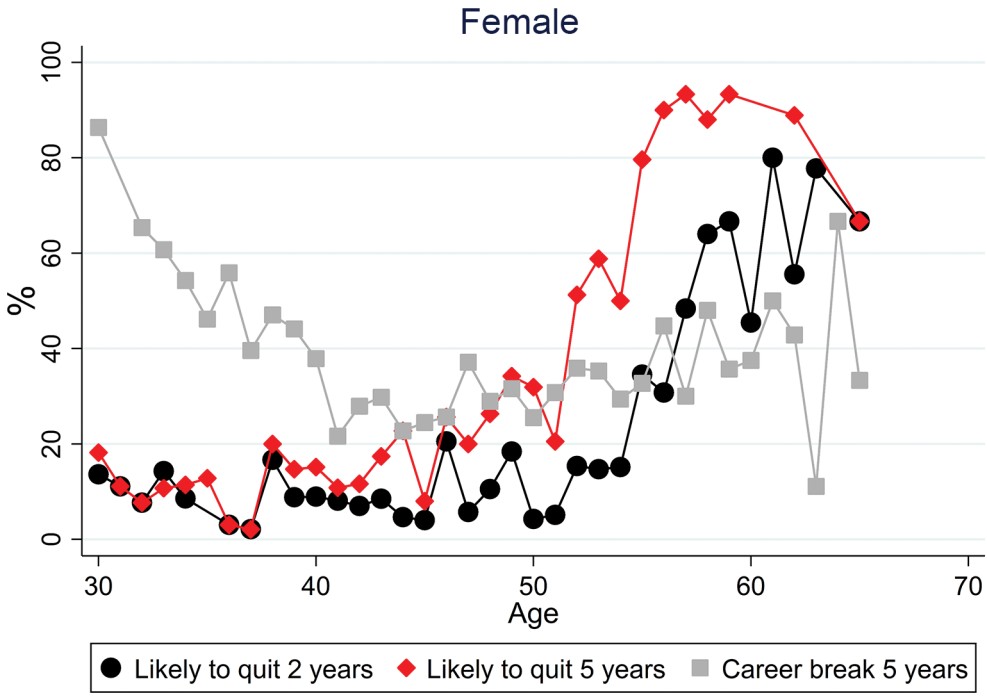

Female

**Figure 1** Career intentions to quit direct patient care by gender and age.

to report intentions to leave direct patient care/reduce hours/take a career break than those with 'high' or 'neither low nor high' morale (and in the case of leaving patient care in 2 years, more likely than any other morale category) but less likely to intend to leave/reduce hours/ take a career break than those with 'very low' morale. This U-shaped relationship largely disappears however after adjustment, and it is those GPs with 'very low' morale who are most likely to report intending to leave direct patient care, reduce hours in patient care or to take a

career break. Moreover, the change in odds between 'low' and 'very low' morale is particularly strong. Table 5 shows the results of the logistic regression analysis modelling of factors associated with low or very low morale. Only age and practice role show any evidence of being associated with reported morale (p<0.001 for both). Those aged 50–54 years old are most likely to report low morale. In respect of role, GP partners are the most likely to report low morale, followed by salaried GPs and with locums and other GPs the least likely groups to report low morale.

**Table 3** Unadjusted associations between career intentions and general practitioner (GP) attributes

| | | How likely is it that you will permanently leave direct patient care within the next 2 years? | | How likely is it that you will permanently leave direct patient care within the next 5 years? | | How likely is it that you will reduce your weekly average hours spent in direct patient care within the next 5 years? | | How likely is it that you will take a career break (or another career break) within the next 5 years? | |
| --- | --- | --- | --- | --- | --- | --- | --- | --- | --- |
| | | OR* (95% CI) | p | OR* (95% CI) | p | OR* (95% CI) | p | OR* (95% CI) | p |
| Gender | Male | Reference | <0.001 | Reference | <0.001 | Reference | <0.001 | Reference | 0.046 |
| | Female | 0.51 (0.41 to 0.63) | | 0.55 (0.46 to 0.66) | | 0.48 (0.41 to 0.58) | | 1.20 (1.00 to 1.44) | |
| Age | <40 | 0.75 (0.48 to 1.16) | <0.001 | 0.49 (0.34 to 0.70) | <0.001 | 1.24 (0.98 to 1.56) | <0.001 | 2.34 (1.83 to 2.98) | <0.001 |
| | 40–49 | Reference | | Reference | | Reference | | Reference | |
| | 50–54 | 1.65 (1.12 to 2.43) | | 4.00 (3.02 to 5.28) | | 2.04 (1.58 to 2.63) | | 1.24 (0.95 to 1.63) | |
| | 55–59 | 9.32 (6.72 to 12.92) | | 30.04 (21.04 to 42.90) | | 5.00 (3.74 to 6.69) | | 1.59 (1.22 to 2.08) | |
| | 60–69 | 22.22 (14.43 to 34.22) | | 91.60 (41.88 to 200.36) | | 7.38 (4.58 to 11.88) | | 1.61 (1.11 to 2.35) | |
| | 70+ | 45.86 (12.84 to 163.84) | | † | | 3.86 (1.23 to 12.10) | | 1.55 (0.56 to 4.32) | |
| Country of qualification | UK/Ireland | Reference | 0.450 | Reference | 0.132 | Reference | 0.921 | Reference | 0.470 |
| | Europe | 0.56 (0.27 to 1.14) | | 0.60 (0.35 to 1.02) | | 0.85 (0.52 to 1.39) | | 0.65 (0.37 to 1.12) | |
| | South Asia | 0.88 (0.29 to 2.66) | | 0.43 (0.14 to 1.32) | | 1.10 (0.45 to 2.70) | | 0.95 (0.38 to 2.39) | |
| | Other | 1.05 (0.47 to 2.32) | | 0.98 (0.51 to 1.90) | | 0.95 (0.50 to 1.80) | | 0.88 (0.45 to 1.73) | |
| Ethnic group | White | Reference | 0.300 | Reference | 0.032 | Reference | 0.581 | Reference | 0.610 |
| | Mixed | † | | 0.60 (0.26 to 1.36) | | 0.74 (0.35 to 1.57) | | 1.57 (0.74 to 3.31) | |
| | Asian | 0.87 (0.48 to 1.57) | | 0.43 (0.25 to 0.76) | | 1.27 (0.78 to 2.05) | | 1.00 (0.61 to 1.63) | |
| | Black | 3.59 (0.89 to 14.41) | | 1.49 (0.37 to 5.99) | | 2.23 (0.45 to 11.07) | | 0.60 (0.12 to 2.99) | |
| | Other | 0.77 (0.22 to 2.69) | | 0.68 (0.24 to 1.96) | | 1.36 (0.50 to 3.70) | | 1.61 (0.62 to 4.18) | |
| Role/position | GP partner | Reference | <0.001 | Reference | <0.001 | Reference | <0.001 | Reference | <0.001 |
| | Salaried GP | 0.61 (0.46 to 0.83) | | 0.43 (0.34 to 0.55) | | 0.63 (0.51 to 0.78) | | 1.25 (1.00 to 1.56) | |
| | Locum GP | 1.58 (1.18 to 2.11) | | 1.08 (0.82 to 1.41) | | 1.34 (1.02 to 1.76) | | 1.72 (1.32 to 2.24) | |
| | Other | 1.12 (0.61 to 2.08) | | 0.72 (0.40 to 1.26) | | 0.79 (0.47 to 1.34) | | 0.85 (0.48 to 1.52) | |
| Reported morale | Very low | 0.99 (0.45 to 2.17) | <0.001 | 1.85 (0.81 to 4.19) | <0.001 | 2.58 (1.19 to 5.59) | <0.001 | 2.24 (1.01 to 4.95) | <0.001 |
| | Low | 0.42 (0.19 to 0.91) | | 0.78 (0.35 to 1.75) | | 1.31 (0.62 to 2.77) | | 1.14 (0.52 to 2.49) | |
| | Neither low/high | 0.30 (0.14 to 0.65) | | 0.62 (0.28 to 1.40) | | 0.82 (0.39 to 1.73) | | 0.81 (0.37 to 1.78) | |
| | High | 0.40 (0.18 to 0.90) | | 0.62 (0.27 to 1.42) | | 0.54 (0.25 to 1.16) | | 0.55 (0.24 to 1.24) | |
| | Very high | Reference | | Reference | | Reference | | Reference | |

*ORs >1 indicate category is more likely to leave direct patient care/reduce hours/take a career break than the reference category.

†This group has been excluded from the analysis due to perfect prediction, that is, all over 70 year olds reported being likely to leave direct patient in 5 years, and no mixed ethnicity GPs reported being likely to leave direct patient care in 2 years.

**Table 4**  Adjusted associations between career intentions and general practitioner (GP) attributes

| | | How likely is it that you will permanently leave direct patient care within the next 2 years? | | How likely is it that you will permanently leave direct patient care within the next 5 years? | | How likely is it that you will reduce your weekly average hours spent in direct patient care within the next 5 years? | | How likely is it that you will take a career break (or another career break) within the next 5 years? | |
|---|---|---|---|---|---|---|---|---|---|
| | | OR* (95% CI) | p | OR* (95% CI) | p | OR* (95% CI) | p | OR* (95% CI) | p |
| Gender | Male | Reference | 0.181 | Reference | 0.810 | Reference | <0.001 | Reference | 0.019 |
| | Female | 0.83 (0.64 to 1.09) | | 1.03 (0.80 to 1.33) | | 0.58 (0.47 to 0.71) | | 1.27 (1.04 to 1.55) | |
| Age | <40 | 0.89 (0.56 to 1.41) | <0.001 | 0.58 (0.40 to 0.85) | <0.001 | 1.47 (1.14 to 1.90) | <0.001 | 2.63 (2.02 to 3.42) | <0.001 |
| | 40–49 | Reference | | Reference | | Reference | | Reference | |
| | 50–54 | 1.70 (1.14 to 2.55) | | 4.73 (3.49 to 6.41) | | 2.11 (1.61 to 2.77) | | 1.31 (0.98 to 1.73) | |
| | 55–59 | 11.69 (8.18 to 16.72) | | 44.15 (29.77 to 65.48) | | 5.62 (4.12 to 7.66) | | 1.79 (1.35 to 2.38) | |
| | 60–69 | 36.87 (22.53 to 60.34) | | 194.43 (84.32 to 448.34) | | 9.58 (5.76 to 15.95) | | 2.03 (1.35 to 3.04) | |
| | 70+ | 110.61 (25.85 to 473.26) | | † | | 4.93 (1.47 to 16.55) | | 1.83 (0.61 to 5.48) | |
| Country of qualification | UK/Ireland | Reference | 0.193 | Reference | 0.048 | Reference | 0.829 | Reference | 0.738 |
| | Europe | 0.60 (0.27 to 1.37) | | 0.48 (0.23 to 0.99) | | 0.98 (0.57 to 1.70) | | 0.73 (0.41 to 1.29) | |
| | South Asia | 0.21 (0.04 to 1.15) | | 0.13 (0.02 to 0.97) | | 0.61 (0.19 to 1.96) | | 0.98 (0.31 to 3.05) | |
| | Other | 0.83 (0.30 to 2.31) | | 0.94 (0.39 to 2.25) | | 0.83 (0.40 to 1.71) | | 1.08 (0.53 to 2.22) | |
| Ethnic group | White | Reference | 0.109 | Reference | 0.142 | Reference | 0.209 | Reference | 0.483 |
| | Mixed | † | | 2.84 (1.15 to 7.04) | | 1.32 (0.60 to 2.92) | | 1.74 (0.79 to 3.81) | |
| | Asian | 1.58 (0.70 to 3.54) | | 0.84 (0.38 to 1.86) | | 1.89 (1.04 to 3.46) | | 0.87 (0.48 to 1.57) | |
| | Black | 6.07 (1.00 to 36.99) | | 2.26 (0.29 to 17.71) | | 2.89 (0.48 to 17.28) | | 0.54 (0.10 to 2.87) | |
| | Other | 0.45 (0.10 to 2.16) | | 0.45 (0.08 to 2.43) | | 1.17 (0.39 to 3.45) | | 1.57 (0.56 to 4.41) | |
| Role/position | GP partner | Reference | 0.177 | Reference | 0.153 | Reference | 0.005 | Reference | <0.001 |
| | Salaried GP | 1.10 (0.76 to 1.59) | | 0.81 (0.58 to 1.14) | | 0.97 (0.75 to 1.24) | | 1.15 (0.89 to 1.47) | |
| | Locum GP | 1.55 (1.04 to 2.30) | | 1.37 (0.91 to 2.07) | | 1.74 (1.26 to 2.39) | | 2.09 (1.55 to 2.81) | |
| | Other | 0.91 (0.39 to 2.15) | | 0.73 (0.29 to 1.86) | | 1.19 (0.64 to 2.20) | | 1.15 (0.62 to 2.13) | |
| Reported morale | Very low | 5.77 (1.96 to 17.00) | <0.001 | 13.07 (3.72 to 45.88) | <0.001 | 6.74 (2.76 to 16.47) | <0.001 | 3.49 (1.49 to 8.14) | <0.001 |
| | Low | 1.87 (0.65 to 5.38) | | 3.89 (1.13 to 13.34) | | 3.27 (1.38 to 7.74) | | 1.50 (0.66 to 3.45) | |
| | Neither low/high | 0.86 (0.30 to 2.47) | | 2.15 (0.63 to 7.39) | | 1.75 (0.74 to 4.14) | | 0.93 (0.40 to 2.12) | |
| | High | 0.80 (0.27 to 2.39) | | 1.29 (0.36 to 4.60) | | 0.86 (0.36 to 2.08) | | 0.59 (0.25 to 1.40) | |
| | Very high | Reference | | Reference | | Reference | | Reference | |

Adjustment was made for all factors shown in table.

*ORs >1 indicate category is more likely to leave direct patient care/reduce hours/take a career break than the reference category.

†This group has been excluded from the analysis due to perfect prediction, that is, all over 70 year olds reported being likely to leave direct patient in 5 years and no mixed ethnicity GPs reported being likely to leave direct patient care in 2 years.

**Table 5** Unadjusted and adjusted associations between self-reported low or very low morale and general practitioner (GP) attributes

| | | Unadjusted | | Adjusted | |
|---|---|---|---|---|---|
| | | OR* (95% CI) | p | OR* (95% CI) | p |
| Gender | Male | Reference | 0.134 | Reference | 0.231 |
| | Female | 0.87 (0.72 to 1.05) | | 0.90 (0.76 to 1.07) | |
| Age | <40 | 0.73 (0.57 to 0.93) | <0.001 | 0.69 (0.55 to 0.88) | <0.001 |
| | 40–49 | Reference | | Reference | |
| | 50–54 | 1.06 (0.81 to 1.37) | | 1.10 (0.85 to 1.43) | |
| | 55–59 | 0.85 (0.66 to 1.10) | | 0.92 (0.71 to 1.18) | |
| | 60–69 | 0.38 (0.26 to 0.57) | | 0.33 (0.23 to 0.48) | |
| | 70+ | 0.07 (0.01 to 0.56) | | 0.04 (0.01 to 0.32) | |
| Country of qualification | UK/Ireland | Reference | 0.157 | Reference | 0.194 |
| | Europe | 0.74 (0.45 to 1.23) | | 0.73 (0.45 to 1.19) | |
| | South Asia | 0.97 (0.32 to 3.01) | | 0.97 (0.40 to 2.36) | |
| | Other | 0.50 (0.25 to 0.98) | | 0.55 (0.29 to 1.06) | |
| Ethnic group | White | Reference | 0.534 | Reference | 0.721 |
| | Mixed | 0.82 (0.38 to 1.77) | | 0.82 (0.39 to 1.72) | |
| | Asian | 1.59 (0.89 to 2.84) | | 1.31 (0.81 to 2.12) | |
| | Black | 1.64 (0.37 to 7.21) | | 0.82 (0.20 to 3.28) | |
| | Other | 1.01 (0.36 to 2.80) | | 0.73 (0.28 to 1.89) | |
| Role/position | GP partner | Reference | <0.001 | Reference | <0.001 |
| | Salaried GP | 0.75 (0.59 to 0.94) | | 0.67 (0.54 to 0.84) | |
| | Locum GP | 0.40 (0.30 to 0.53) | | 0.33 (0.25 to 0.43) | |
| | Other | 0.22 (0.12 to 0.41) | | 0.20 (0.11 to 0.36) | |

Adjustment was made for all factors shown in table.
*ORs >1 category is more likely to report low or very low morale than the reference category.

Supplementary analyses exploring the contribution of current working patterns to any association with intentions to reduce hours spent in direct patient care did explain some of the difference between genders. However, even after adjustment for current working patterns, female GPs were still substantially less likely to report intentions to reduce hours spent in direct patient care than their male counterparts (not shown).

## DISCUSSION
### Main findings
A high proportion of GPs—around two in every five—currently working within direct patient care in South West England reported an intention to permanently quit direct patient care within the next 5 years, this being one in five within the next 2 years. Further depletion of the GP workforce in this region of England through reduction of weekly average hours or through taking a career break within the next 5 years also emerged as an impending risk. Overall, 7 out of every 10 GPs in this region reported a career intention which, if implemented, would adversely impact the GP workforce capacity in South West England through GPs leaving direct patient care, reducing hours

spent in direct patient care or by taking a career break within the next 5 years.

Older age is highly predictive of a GP's intentions to permanently quit direct patient care and to reduce hours. The intention to quit was independent of the GP's role (partner, salaried or locum) and gender. The gender differences observed among younger GPs in respect of intended career breaks is a particular issue where, as currently exists, 69% of the GP trainee workforce are female,[17] and where a substantial proportion of younger women returning to clinical care do so on the basis of reduced hours[1] and where doctors who wish to return to clinical practice may face significant bureaucratic obstacles in doing so.[18] The decision to reduce hours or to take a career break varied with employment status, with locum GPs being most likely to report intentions to reduce hours or to take a career break. In a situation where GPs who are not partners in a practice do in fact leave direct patient care, the system of care appears vulnerable, with added strain likely to be placed on GPs who are partners in their practice.

A key issue is whether the intention to leave is translated into reality, and few studies have explored this important

question. However, Hann *et al*[19] explored this question among nearly 1200 UK GPs followed up over 5 years, and identified a strong relationship between a stated intention to leave patient care and actually doing so, reflected in a 4.5-fold difference in odds of leaving between those with no stated intention to leave patient care when compared with those reporting a high intention to do so.

The majority of responding GPs in our study reported low morale, with less than 15% reporting high morale. GP morale was identified as an important predictor of future career intentions, particularly when morale was very low.

### Strengths and limitations of the study

By ensuring alignment of wording with other surveys of similar intent allows the opportunity for comparing and contrasting of results. Recent high-quality surveys support our estimate of the proportion of GPs intending to leave general practice within the next 5 years.[14 20–22] Unlike most other similar surveys which tend to survey samples of GPs,[7 23] we undertook a census survey of all GPs currently eligible to provide patient care across a large area, providing a cross-sectional overview of the quit intentions of all GPs in South West England. The response rate was high (67%), reflecting both rigorous planning and implementation of the survey and interest among GPs in the subject matter of workforce challenges. Further, we have reduced the potential for non-response bias by weighting our estimates to account for differential response rates by age, gender and practice role. The overall programme of work, of which this survey is one element, includes the planned development of a predictive risk modelling tool to enumerate risk of future workforce undersupply at local levels, thus offering the potential of providing data that will be of value to healthcare planners and policy makers. In addition, our study relates to a key current area of national interest and concern.

The study has a number of limitations. The survey was conducted among GPs in a single region of England which may not be entirely typical of England as a whole, for example in respect of population mix and practice setting.[24] The region is often informally regarded as having desirable living and working conditions.[25] It seems likely that other regions of England, with more inner-city areas with higher levels of deprivation, are likely to face even greater challenges in respect of the GP workforce than in the findings we have presented here. A further limitation is that the research was undertaken within a rapidly changing GP workforce policy environment, for example, the implementation in 2015–2016 of a 10-point plan agreed between authoritative UK health, governmental and professional bodies concerned about GP workforce issues.[26] Finally, this survey was cross-sectional, rather than longitudinal in design, and so we are unable to report on the actual translation of GPs' reported career intentions into actual career decisions, although previous work has demonstrated intention to quit to be a strong predictor of actually leaving.[19]

### Implications for research

Our survey is part of a multimethod suite of work commissioned by the NHS, seeking to inform evidence-based policies and strategies in a key area of national concern and which will involve stakeholder consultations with policy makers who have oversight of GP workforce planning.[16] Future research should include assessment of the predictive validity of GPs' reported career intentions by longitudinal follow-up of GPs, investigation of how reducing hours or better flexible working arrangements may influence GP retention and undertaking research exploring the potential for extended clinical roles among important allied healthcare professionals in the clinical workforce. Such groups might include nurses, physiotherapists and pharmacists, and developing such extended roles for these professionals within the primary care team may offer a rapid and potentially effective contribution to the alleviation of current GP workforce pressures. Research into the determinants and impact of working reduced hours should also encompass other aspects of flexible working arrangements in general practice, and how such arrangements may improve job satisfaction and work–life balance and thereby potentially increase GP retention in the longer term. This work also provides insight into which groups of GPs might best be targeted with interventions aiming to improve workforce retention.

### Implications for policy and practice

Concerns regarding the GP workforce are now recognised widely by authoritative bodies in the UK, leading to recent policy statements and initiatives[26] and to the establishment of a Primary Care Workforce Commission by Health Education England.[4] Our research can inform this policy agenda by providing an estimate of the likely proportion of South West GPs intending to leave direct patient care within the next 5 years through permanently quitting patient care or by taking a career break. Our findings are likely to be of relevance in other areas of England that may face even greater challenges in respect of the GP workforce. Retaining the direct patient care capacity of established GPs represents an immediate challenge for healthcare planners. Recognising and addressing issues of low GP morale is an area of importance, and addressing the issue may improve how GPs feel about their work and potentially be associated with improvements in patients' experience of the care received.[27] In the face of problems recruiting to general practice[28] and the wider workforce problems, the UK government recently announced an increase in the number of training places for GPs[26] and an increase in the number of medical training places in the UK.[29] But these medium-term to long-term solutions do not address the immediate workforce crisis. In contrast, policy initiatives to support the retention of the existing GP workforce, the rapid development of and support for new models of care and the implementation of policies aimed at alleviating workload pressures for GPs are potential short-term to medium-term solutions

which need to be urgently considered. Developing and implementing relevant policies and strategies to achieve those ambitions will be important for the alleviation of the immediate pressures faced by the GP workforce and the populations of patients they serve.

## CONCLUSIONS

Healthcare in England faces imminent challenges in respect of GP workforce capacity. This survey identifies the magnitude of the problem in South West England and highlights the important role of GP morale as one important factor contributing to that challenge. Acknowledgement of the magnitude of the problems is urgently required, along with implementation and monitoring of relevant policy and strategy. Failure to do so will risk serious adverse effects on the capacity and ability of the NHS to provide effective primary care to the UK population.

**Contributors** EF contributed to the study design, data collection, analysis and writing of the paper. JLC contributed to study design, analysis and writing of the paper. RA, SR, CS, SGD, AS and FCW contributed to the study design and writing of the paper. GAA contributed to analysis and writing of the paper. All authors read and approved the final paper. JLC is the guarantor of the paper.

**Competing interests** None declared.

**Ethics approval** University of Exeter Medical School Research Ethics Committee.

**Provenance and peer review** Not commissioned; externally peer reviewed.

**Data sharing statement** Requests for statistical code and dataset can be made to the corresponding author at john.campbell@exeter.ac.uk. The dataset will be made available via a publicly accessible repository on publication. All authors, external and internal, had full access to all of the data (including statistical reports and tables) in the study and can take responsibility for the integrity of the data and the accuracy of the data analysis.

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
