## [Reviewer comments · BMJ Open]

ARTICLE DETAILS

TITLE (PROVISIONAL)	Quitting patient care and career break intentions among general practitioners in South West England: findings of a census survey of general practitioners.
AUTHORS	Fletcher, Emily; Abel, Gary; Anderson, Rob; Richards, Suzanne; Salisbury, Chris; Dean, Sarah; Sansom, Anna; Warren, Fiona; Campbell, John

VERSION 1 - REVIEW

REVIEWER	Dr Sharon Wiener-Ogilvie NHS education Scotland
REVIEW RETURNED	31-Jan-2017

GENERAL COMMENTS	This is a well written paper. Although there is already existing evidence relating to the retention of GPs, this study provide a more thorough and detailed picture albeit in a particular geographical area of England.
--

REVIEWER	Jennifer Cleland University of Aberdeen, UK
REVIEW RETURNED	02-Mar-2017

GENERAL COMMENTS	I can rarely resist suggesting ways to revise a paper, but in this case I cannot find anything that can improved upon. This is a very through analysis addressing a very important question. My only suggestion is to repeat the study in other regions!
--

VERSION 1 – AUTHOR RESPONSE

Very many thanks for this. We appreciate the two short but extremely positive responses. The comments by Professor Clelland are already specifically addressed in the limitations section of the manuscript.